**Data Availability Statement:** All relevant data are uploaded to Our World in Data

## RESEARCH ARTICLE

# The Global Health Security Index is not predictive of coronavirus pandemic responses among Organization for Economic Cooperation and Development countries

Enoch J. Abbey[1], Banda A. A. Khalifa[2], Modupe O. Oduwole[1,2], Samuel K. Ayeh[1], Richard D. Nudotor[3], Emmanuella L. Salia[2,4], Oluwatobi Lasisi[5], Seth Bennett[6], Hasiya E. Yusuf[4], Allison L. Agwu[1,4☯‡]*, Petros C. Karakousis[1,7☯‡]*

1 Department of Medicine, Johns Hopkins School of Medicine, Baltimore, Maryland, United States of America, 2 Department of Epidemiology, Johns Hopkins Bloomberg School of Public Health, Baltimore, Maryland, United States of America, 3 Department of Surgery, Johns Hopkins School of Medicine, Baltimore, Maryland, United States of America, 4 Department of Pediatrics, Johns Hopkins School of Medicine, Baltimore, Maryland, United States of America, 5 Wayne State University School of Medicine, Detroit, Michigan, United States of America, 6 CTI Clinical Trial and Consulting, Covington, Kentucky, United States of America, 7 Department of International Health, Johns Hopkins Bloomberg School of Public Health, Baltimore, Maryland, United States of America

☯ These authors contributed equally to this work.
‡ These authors are joint senior authors on this work.
* petros@jhmi.edu (PCK); ageorg10@jhmi.edu (ALA)

## Abstract

The ongoing COVID-19 pandemic has devastated many countries with ripple effects felt in various sectors of the global economy. In November 2019, the Global Health Security (GHS) Index was released as the first detailed assessment and benchmarking of 195 countries to prevent, detect, and respond to infectious disease threats. This paper presents the first comparison of Organization for Economic Cooperation and Development OECD countries' performance during the pandemic, with the pre-COVID-19 pandemic preparedness as determined by the GHS Index. Using a rank-based analysis, four indices were compared between select countries, including total cases, total deaths, recovery rate, and total tests performed, all standardized for comparison. Our findings suggest a discrepancy between the GHS index rating and the actual performance of countries during this pandemic, with an overestimation of the preparedness of some countries scoring highly on the GHS index and underestimation of the preparedness of other countries with relatively lower scores on the GHS index.

## Background

The outbreak of the novel coronavirus disease (COVID-19), caused by severe acute respiratory syndrome coronavirus 2 (SARS-CoV-2), has been described as the greatest global health threat of the century. First discovered in Wuhan, China in late 2019, the infection was declared a

(https://ourworldindata.org/coronavirus-testing)
and GitHub (https://github.com/CSSEGISandData/
COVID-19).

**Funding:** The National Institute of Allergy and
Infectious Diseases (NIAID) supported this study in
the form of a grant awarded to PCK
(K24AI143447). The funder had no role in study
design, data collection and analysis, decision to
publish, or preparation of the manuscript.

**Competing interests:** The authors have declared
that no competing interests exist.

pandemic by the World Health Organization in March 2020. As of June 29, 2020, there are
over 10·2 million confirmed cases of COVID-19 globally [1]. Despite remarkable efforts by
global health agencies and governments, the number of new infections is projected to grow in
the coming months [1]. The pandemic has also claimed many lives, with over 450,000 deaths
recorded globally [1].

The COVID-19 pandemic has significantly impacted healthcare systems with rippling
effects in various sectors of the global economy. Frontline healthcare workers who come in
direct contact with patients are at increased risk of being infected with SARS-CoV-2. Overall
healthcare delivery to the general population has been affected by the staggering increase in
the demand for medical supplies, reduced in-person medical visits, and shortages of medical
protective gear [2]. In addition to the health implications, efforts to control the pandemic have
led to massive lockdowns (stay-at-home orders) and social distancing guidelines, causing sub-
stantial financial losses for individuals, businesses, governments, and threatened recessions in
some countries [2]. In many countries, the capital market sectors have been affected, and there
have been disruptions to the global supply chain.

The world is intricately interconnected through the movement of people and trade across
borders, facilitating the spread of infectious diseases, and posing serious risks to global health.
Therefore, it is imperative that countries are able to promptly identify and respond to cata-
strophic public health emergencies. The Global Health Security (GHS) index is the first com-
prehensive assessment of countries' preparedness of countries for outbreaks like COVID-19.
The GHS index project was conducted by the Nuclear Threat Initiative (NTI), the Johns Hop-
kins Center for Health Security (JHU), and the Economist Intelligence Unit (EIU) to thor-
oughly evaluate the health security and related capabilities of 195 countries that are Parties to
the International Health Regulations (2005) [3].

In 2019, a panel of 21 experts from 13 countries developed an elaborate and comprehensive
framework to assess a country's ability to avert and mitigate outbreaks. The international
panel of experts scored each of the 195 countries and classified them as 'most prepared'
(score $\geq$ 66.7), 'more prepared' (33.4 to 66.6) and 'least prepared' (0 to 33.3) [3]. The GHS
index is based on 6 categories, 34 indicators, and 85 sub-indicators. The six categories com-
prise prevention, detection and reporting, rapid response, health system, compliance with
international norms, and risk environment [3]. Findings from the evaluation revealed that
none of the countries is fully prepared for an infectious disease outbreak or a pandemic, with a
global average score of ~40 out of 100 [3], underscoring critical gaps in outbreak and epi-
demic/pandemic preparedness that must be urgently addressed. The average GHS index score
among the 60 high-income countries was ~50, indicating a weak collective pandemic pre-
paredness. The United States (US), United Kingdom (UK), Netherlands, Australia, and Can-
ada ranked in the top 5 countries on the GHS index, with scores of 83.5, 77.9, 75.6, 75.5, and
75.3, respectively. Recently, the GHS index has been used to assess the preparedness of coun-
tries to handle the novel COVID-19 pandemic and thus far demonstrates suboptimal levels of
preparedness for all countries assessed. However, the top 5 countries as ranked by the GHS
index are among the worst-hit countries by COVID-19, with a high number of cases and mor-
talities [1].

The Organization for Economic Cooperation and Development (OECD) is currently a
37-member association, which is in the process of adding Costa Rica as its 38th nation [4, 5]. It
comprises wealthy nations that have both high energy consumption and high gross domestic
products. Headquartered in France, members jointly contribute to 80% of global trade and
investment [6]. The OECD is influential in supporting non-member nations with resources to
improve economies, while also monitoring economies and the ability of states to fight poverty.

The objective of this study is to evaluate the utility of the GHS index in predicting the current responses of 36/37 OECD countries, for which data are available, to the COVID-19 pandemic.

## Methodology

### Study population

Participating countries used for this analysis include OECD member countries [7], each of which was assessed for inclusion. COVID-19 data are reported on a daily basis for all countries, with the exception of tests/ thousand people, which are available on a weekly basis for the following countries: Netherlands, Spain, Sweden, Ireland, and Germany. France was excluded from the analysis due to absent data on tests/ thousand people. Thirty-six countries were included in the analysis.

### Data collection

We collected country-level data on the preparedness to prevent, detect and respond to infectious disease threats using the Global Health Security (GHS) index available from https://www.ghsindex.org/ [8]. Data relating to COVID-19 cases, deaths, recoveries, and number of tests were obtained from https://ourworldindata.org/coronavirus-testing (a collaborative effort between the researchers at the University of Oxford and a non-profit organization, Global Change Data Lab) [9] and https://github.com/CSSEGISandData/COVID-19 (a COVID-19 data repository by the Center for Systems Science and Engineering at Johns Hopkins University) [10] for May 15, 2020.

### Statistical analysis

Our variables of interest included both the raw and standardized total number of COVID-19 cases, deaths, tests performed, and recoveries as of May 15, 2020. We estimated a modified recovery rate as the ratio of the total number of recoveries to the total number of cases for each country (total recoveries/ total cases * 100%), as timelines and dates for the first incident cases were not readily available. The 36 countries were rank-ordered based on the total number of cases/ million and the total number of deaths/ million from the lowest to the highest, with each assigned a score ranging from 1 to 36. The 36 countries were also rank-ordered based on the total tests per thousand people and recovery rate, with the highest recovery rate and tests per thousand assigned a score of 1, and the lowest recovery rate and tests per thousand assigned a rating of 36. The cumulative score involving all four variables of interest were equally weighted for each criterion. This was done by calculating the average rank for the cumulative score and then to contrive their final multi-criteria rankings. Using this approach, the lowest cumulative score has the lowest multi-criteria rank and the highest aggregate score has the highest multi-criteria rank. A lower rating reflects relatively better performance, and Spearman's rank correlation was determined between the GHS index and the commutative ranking. All statistical analyses were performed using STATA (Statistical Data Analysis Package version 16.0 IC, College Station, TX–USA).

## Results

The US ranks highest in preparedness, with a score of 83.5, a five-point difference from the next country, the UK. The US ranks highest in four out of six areas: prevention, detection, health system capacity, and compliance with international norms. Luxembourg ranks at the

**Table 1. Country ranking and score based on the Global Health Survey index.**

| Country (ranked) | Overall Score | Prevention of the emergence or release of pathogens | Early detection & reporting for epidemics of potential international concern | Rapid response to and mitigation of the spread of an epidemic | Sufficient & robust health system to treat the sick and protect health workers | Commitments to improving national capacity, financing and adherence to norms | Overall risk environment and country vulnerability to biological threats |
|---|---|---|---|---|---|---|---|
| | Rank (score) | Rank (score) | Rank (score) | Rank (score) | Rank (score) | Rank (score) | Rank (score) |
| United States | 1 (83.5) | 1 (83.1) | 1 (98.2) | 2 (79.7) | 1 (73.8) | 1 (85.3) | 19 (78.2) |
| United Kingdom | 2 (77.9) | 10 (68.3) | 6 (87.3) | 1 (91.9) | 11 (59.8) | 2 (81.2) | 26 (74.7) |
| Netherlands | 3 (75.6) | 4 (73.7) | 7 (86.0) | 4 (79.1) | 3 (70.2) | 32 (61.1) | 12 (81.7) |
| Australia | 4 (75.5) | 8 (68.9) | 2 (97.3) | 10 (65.9) | 6 (63.5) | 3 (77.0) | 18 (79.4) |
| Canada | 5 (75.3) | 7 (70.0) | 4 (96.4) | 17 (60.7) | 4 (67.7) | 5 (74.7) | 10 (82.7) |
| Sweden | 7 (72.1) | 2 (81.1) | 7 (86.0) | 14 (62.8) | 20 (49.3) | 11 (71.3) | 6 (84.5) |
| Denmark | 8 (70.4) | 5 (72.9) | 7 (86.0) | 19 (58.4) | 5 (63.8) | 28 (62.6) | 17 (80.3) |
| South Korea | 9 (70.2) | 19 (57.3) | 5 (92.1) | 6 (71.5) | 13 (58.7) | 23 (64.3) | 27 (74.1) |
| Finland | 10 (68.7) | 9 (68.5) | 45 (61.6) | 7 (69.2) | 9 (60.8) | 4 (75.4) | 14 (81.1) |
| Slovenia | 12 (67.2) | 12 (67.0) | 27 (73.7) | 12 (63.3) | 18 (54.9) | 8 (72.1) | 29 (73.7) |
| Switzerland | 13 (67.0) | 34 (52.7) | 48 (59.1) | 3 (79.3) | 7 (62.5) | 18 (65.6) | 3 (86.2) |
| Germany | 14 (66.0) | 13 (66.5) | 10 (84.6) | 28 (54.8) | 22 (48.2) | 29 (61.9) | 11 (82.3) |
| Spain | 15 (65.9) | 32 (52.9) | 11 (83.0) | 15 (61.9) | 12 (59.6) | 32 (61.1) | 24 (77.1) |
| Norway | 16 (64.6) | 11 (68.2) | 49 (58.6) | 20 (58.2) | 14 (58.5) | 22 (64.4) | 2 (87.1) |
| Latvia | 17 (62.9) | 25 (56.0) | 2 (97.3) | 29 (54.7) | 23 (47.3) | 79 (51.1) | 48 (67.2) |
| Belgium | 19 (61.0) | 15 (63.5) | 42 (62.5) | 53 (47.3) | 10 (60.5) | 38 (59.7) | 19 (78.2) |
| Portugal | 20 (60.3) | 33 (52.8) | 61 (50.5) | 8 (67.7) | 17 (55.0) | 26 (63.0) | 22 (77.3) |
| Japan | 21 (59.8) | 40 (49.3) | 35 (70.1) | 31 (53.6) | 25 (46.6) | 13 (70.0) | 34 (71.7) |
| Ireland | 23 (59.0) | 14 (63.9) | 18 (78.0) | 62 (45.1) | 41 (40.2) | 66 (52.8) | 21 (77.4) |
| Austria | 26 (58.5) | 18 (57.4) | 28 (73.2) | 76 (42.3) | 25 (46.6) | 66 (52.8) | 5 (84.6) |
| Chile | 27 (58.3) | 23 (56.2) | 30 (72.7) | 18 (60.2) | 43 (39.3) | 78 (51.5) | 38 (70.1) |
| Mexico | 28 (57.6) | 49 (45.5) | 32 (71.2) | 39 (50.8) | 24 (46.9) | 6 (73.9) | 89 (57.0) |
| Estonia | 29 (57.0) | 44 (47.6) | 19 (77.6) | 56 (47.0) | 66 (31.6) | 15 (67.6) | 30 (73.3) |
| Italy | 31 (56.2) | 45 (47.5) | 16 (78.5) | 51 (47.5) | 54 (36.8) | 29 (61.9) | 55 (65.5) |
| Poland | 32 (55.4) | 37 (50.9) | 44 (61.7) | 51 (47.5) | 21 (48.9) | 41 (58.9) | 45 (67.9) |
| Lithuania | 33 (55.0) | 59 (43.5) | 13 (81.5) | 107 (33.9) | 63 (34.4) | 8 (72.1) | 46 (67.8) |
| Hungary | 35 (54.0) | 22 (56.4) | 55 (55.5) | 33 (52.2) | 56 (36.6) | 41 (58.9) | 42 (68.2) |
| New Zealand | 35 (54.0) | 27 (55.0) | 107 (36.7) | 21 (58.1) | 32 (45.2) | 39 (59.4) | 23 (77.2) |
| Greece | 37 (53.8) | 28 (54.2) | 17 (78.4) | 66 (44.0) | 50 (37.6) | 92 (49.1) | 80 (58.2) |
| Turkey | 40 (52.4) | 20 (56.9) | 74 (45.6) | 46 (49.0) | 30 (45.7) | 23 (64.3) | 92 (56.5) |
| Czech Republic | 42 (52.0) | 36 (51.1) | 60 (50.7) | 57 (46.6) | 52 (37.4) | 41 (58.9) | 28 (74.0) |
| Slovakia | 52 (47.9) | 30 (53.5) | 70 (46.0) | 105 (34.1) | 48 (37.9) | 66 (52.8) | 36 (71.5) |
| Israel | 54 (47.3) | 54 (44.0) | 58 (52.4) | 84 (39.9) | 37 (42.2) | 138 (41.5) | 41 (68.8) |
| Iceland | 58 (46.3) | 84 (35.3) | 104 (37.2) | 66 (44.0) | 28 (46.4) | 128 (43.2) | 13 (81.2) |
| Colombia | 65 (44.2) | 75 (37.2) | 91 (41.7) | 70 (43.5) | 64 (34.3) | 35 (60.1) | 116 (51.0) |
| Luxembourg | 67 (43.8) | 102 (31.0) | 91 (41.7) | 139 (27.3) | 48 (37.9) | 66 (52.8) | 4 (84.7) |

67th position in the global ranking, with an overall score of 43.8. The 20 highest ranked countries are mostly OECD members, except for Thailand (6th) and Malaysia (18th) (Table 1).

In absolute terms, the US ranks worst globally in terms of the absolute numbers of cases, and deaths since the onset of the pandemic. The US ranks better than only 4 other countries

with respect to the number of cases per million, deaths per million, recovery rate, and tests per thousand. Of the 36 OECD countries, the US ranks 32nd followed by Spain, Sweden, the UK, and the Netherlands, which rank in 33rd to 36th place, respectively (Table 2). New Zealand, which ranks 35th globally based on the GHS index, tops the performance list based on the four variables of interest, followed by Australia, South Korea and Lithuania (tie), which occupy the 4th, 9th, and 33rd positions, respectively, and outperform other countries ranked in the top 10 based on GHS index. Although there are limitations to this approach, the two most prepared countries based on the GHS index perform worse than other OECD countries. New Zealand tops the recovery rate ranking per our definition, followed closely by Iceland and Luxembourg. However, this parameter depends largely on the timing of index cases and whether countries have reached their peak surge of cases, and, as such, has limited utility. Iceland has conducted the most tests per thousand people, while Japan and Australia have the lowest cases and deaths per million, respectively.

Luxembourg ranks 67th globally according to the GHS index and last among the OECD countries analyzed. However, as of May 15, 2020, it had outperformed twenty-one other OECD countries. It has the highest cases per million but also ranks highly with respect to tests performed per thousand population and recovery rate. Countries ranking in the bottom 5 as per the GHS index, including Slovakia, Israel, and Iceland, are in the top 10 OECD countries per their performance against the variables of interest (Fig 1). Conversely, countries ranked in the top five of the GHS index, occupy the bottom eight positions of the commutative ranking. These are the US, UK, Netherlands, and Canada. The various indices and variables used in the ranking of the countries are captured in Table 3. There is a negative moderate relationship between the GHS index ranking and the computed commutative ranks, with a correlation coefficient ($r_s$) of -0.41 and a p-value of 0.013.

## Discussion

Our study's findings indicate a negative correlation between the GHS rankings and the multi-criteria rank we developed based on our COVID-19 performance indicators, highlighting the lack of utility of the GHS index in predicting the response of countries to the COVID-19 pandemic and its impact on them. Specifically, the favorable ranking of the US and UK based on the GHS index vis-à-vis their preparedness against the threat of infectious agents is inconsistent with current data from the COVID-19 pandemic. Thus, the US and UK are among the top 10 countries with the highest number of cases per million people. While Japan and South Korea were ranked 21st and 9th on the GHS index rating [11], they ranked 1st and 2nd in the (lowest) number of cases per million, suggesting successful mechanisms for dealing with the pandemic. Additionally, our multi-criteria ranking system places New Zealand, Australia, and South Korea as the top three countries with successful mechanisms for handling the ongoing pandemic, although they ranked 35th, 4th, and 9th on the GHS index, respectively. These inconsistencies reveal that the GHS index rating may have overestimated the robustness of certain national health care systems and their level of bio-preparedness while underestimating those of others.

The GHS index report revealed significant weaknesses in every country's overall preparedness level, however it is imperative to understand why there is a discrepancy between the GHS index and the actual level of pandemic preparedness among the OECD countries studied [11, 12]. Although the performance of Australia and South Korea were consistent with their ranking on the GHS index, New Zealand was the best-performing country among the OECD countries, raising the question of the reliability of the GHS index rankings. South Korea and other Asian countries have provided swift, effective ways of dealing with the outbreak, perhaps due

**Table 2.  Ranking of OECD countries based on variables of interest.**

| OECD countries ranked on GHS Index Ranking | OECD countries ranked by cases/million (lowest to highest) | OECD countries ranked by deaths/million (lowest to highest) | OECD countries ranked by recovery rate (highest to lowest) | OECD countries ranked by tests/thousand (highest to lowest) | OECD countries ranked by Cumulative Score (lowest to highest) | Average Score | Final Multi-criteria Rank for OECD countries |
|---|---|---|---|---|---|---|---|
| US (1) | Japan (1) | Australia (1) | New Zealand (1) | Iceland (1) | New Zealand (17) | New Zealand (4.3) | New Zealand (1) |
| UK (2) | South Korea (2) | New Zealand (2) | Iceland (2) | Luxembourg (2) | Australia (29) | Australia (7.3) | Australia (2) |
| Netherlands (3) | New Zealand (3) | Slovakia (3) | Luxembourg (3) | Lithuania (3) | South Korea (43) | South Korea (10.8) | South Korea (3) |
| Australia (4) | Greece (4) | South Korea (4) | Australia (4) | Denmark (4) | Lithuania (43) | Lithuania (10.8) | Lithuania (4) |
| Canada (5) | Colombia (5) | Japan (5) | Austria (5) | Portugal (5) | Slovakia (45) | Slovakia (11.3) | Slovakia (5) |
| Sweden (7) | Slovakia (6) | Latvia (6) | South Korea (6) | Israel (6) | Latvia (45) | Latvia (11.3) | Latvia (6) |
| Denmark (8) | Australia (7) | Colombia (7) | Switzerland (7) | Ireland (7) | Iceland (51) | Iceland (12.8) | Iceland (7) |
| South Korea (9) | Mexico (8) | Greece (8) | Germany (8) | Estonia (8) | Israel (54) | Israel (13.5) | Israel (8) |
| Finland (10) | Hungary (9) | Chile (9) | Denmark (9) | Belgium (9) | Denmark (56) | Denmark (14) | Denmark (9) |
| Slovenia (12) | Poland (10) | Lithuania (10) | Ireland (10) | Italy (10) | Japan (60) | Japan (15) | Japan (10) |
| Switzerland (13) | Latvia (11) | Poland (11) | Finland (11) | New Zealand (11) | Austria (62) | Austria (15.5) | Austria (11) |
| Germany (14) | Lithuania (12) | Czech Republic (12) | Slovakia (12) | Latvia (12) | Estonia (63) | Estonia (15.8) | Estonia (12) |
| Spain (15) | Slovenia (13) | Iceland (13) | Israel (13) | Spain (13) | Czech Republic (64) | Czech Republic (16) | Czech Republic (13) |
| Norway (16) | Czech Republic (14) | Israel (14) | Turkey (14) | Norway (14) | Greece (68) | Greece (17) | Greece (14) |
| Latvia (17) | Finland (15) | Mexico (15) | Mexico (15) | Switzerland (15) | Luxembourg (68) | Luxembourg (17) | Luxembourg (15) |
| Belgium (19) | Estonia (16) | Norway (16) | Latvia (16) | Austria (16) | Finland (70) | Finland (17.5) | Finland (16) |
| Portugal (20) | Norway (17) | Hungary (17) | Czech Republic (17) | Australia (17) | Colombia (74) | Colombia (18.5) | Colombia (17) |
| Japan (21) | Turkey (18) | Estonia (18) | Lithuania (18) | Germany (18) | Germany (74) | Germany (18.5) | Germany (18) |
| Ireland (23) | Austria (19) | Turkey (19) | Japan (19) | Slovenia (19) | Mexico (74) | Mexico (18.5) | Mexico (19) |
| Austria (26) | Denmark (20) | Slovenia (20) | Italy (20) | Canada (20) | Poland (76) | Poland (19) | Poland (20) |
| Chile (27) | Israel (21) | Finland (21) | Estonia (21) | Czech Republic (21) | Turkey (77) | Turkey (19.3) | Turkey (21) |
| Mexico (28) | Chile (22) | Austria (22) | Canada (22) | US (22) | Switzerland (79) | Switzerland (19.8) | Switzerland (22) |
| Estonia (29) | Canada (23) | Denmark (23) | Greece (23) | Finland (23) | Ireland (80) | Ireland (20) | Ireland (23) |
| Italy (31) | Germany (24) | Germany (24) | Chile (24) | Slovakia (24) | Slovenia (81) | Slovenia (20.3) | Slovenia (24) |
| Poland (32) | Netherlands (25) | Portugal (25) | Poland (25) | UK (25) | Chile (82) | Chile (20.5) | Chile (25) |
| Lithuania (33) | Portugal (26) | Canada (26) | Hungary (26) | Turkey (26) | Norway (82) | Norway (20.5) | Norway (26) |
| Hungary (35) | Sweden (27) | Luxembourg (27) | Belgium (27) | Chile (27) | Hungary (84) | Hungary (21) | Hungary (27) |
| New Zealand (35) | UK (28) | Switzerland (28) | Colombia (28) | Sweden (28) | Portugal (88) | Portugal (22) | Portugal (28) |
| Greece (37) | Switzerland (29) | US (29) | Slovenia (29) | Netherlands (29) | Canada (91) | Canada (22.8) | Canada (29) |
| Turkey (40) | Italy (30) | Ireland (30) | US (30) | Poland (30) | Italy (94) | Italy (23.5) | Italy (30) |
| Czech Republic (42) | US (31) | Netherlands (31) | Sweden (31) | South Korea (31) | Belgium (104) | Belgium (26) | Belgium (31) |
| Slovakia (52) | Belgium (32) | Sweden (32) | Portugal (32) | Hungary (32) | US (112) | US (28) | US (32) |
| Israel () | Ireland (33) | UK (33) | Spain (33) | Greece (33) | Spain (115) | Spain (28.8) | Spain (33) |
| Iceland (58) | Spain (34) | Italy (34) | UK (34) | Colombia (34) | Sweden (118) | Sweden (29.5) | Sweden (34) |
| Colombia (65) | Iceland (35) | Spain (35) | Norway (35) | Japan (35) | UK (120) | UK (30) | UK (35) |

*(Continued)*

**Table 2.** (Continued)

| OECD countries ranked on GHS Index Ranking | OECD countries ranked by cases/ million (lowest to highest) | OECD countries ranked by deaths/ million (lowest to highest) | OECD countries ranked by recovery rate (highest to lowest) | OECD countries ranked by tests/ thousand (highest to lowest) | OECD countries ranked by Cumulative Score (lowest to highest) | Average Score | Final Multi-criteria Rank for OECD countries |
|---|---|---|---|---|---|---|---|
| **Luxembourg (67)** | Luxembourg (36) | Belgium (36) | Netherlands (36) | Mexico (36) | Netherlands (121) | Netherlands (30.3) | Netherlands (36) |

to their experience with the outbreak of Severe Acute Respiratory Syndrome (SARS) in 2003 and the Middle East Respiratory Syndrome (MERS) in 2015 [13–15]. The lower death rates reported in Australia, New Zealand, Slovakia, South Korea, and Japan may also be due to extensive testing, rapid surveillance, and effectively enforced quarantine and isolation mechanisms.

The discrepancies between the GHS index rankings and the actual response to the COVID-19 pandemic based on our indicators among the OECD countries may also highlight some deficits in the weighting of categories and the sources of data utilized by the expert panel [14]. Importantly, the GHS index expert panel did not directly engage authorities responsible for emergency preparedness in their respective countries and other key stakeholders. Instead, the panel evaluated information provided by each country; this methodology has the potential to obscure crucial weaknesses in a country's capacity to confront outbreaks. Although the US scored the highest (98.2) in the category of early detection and reporting of epidemics, our

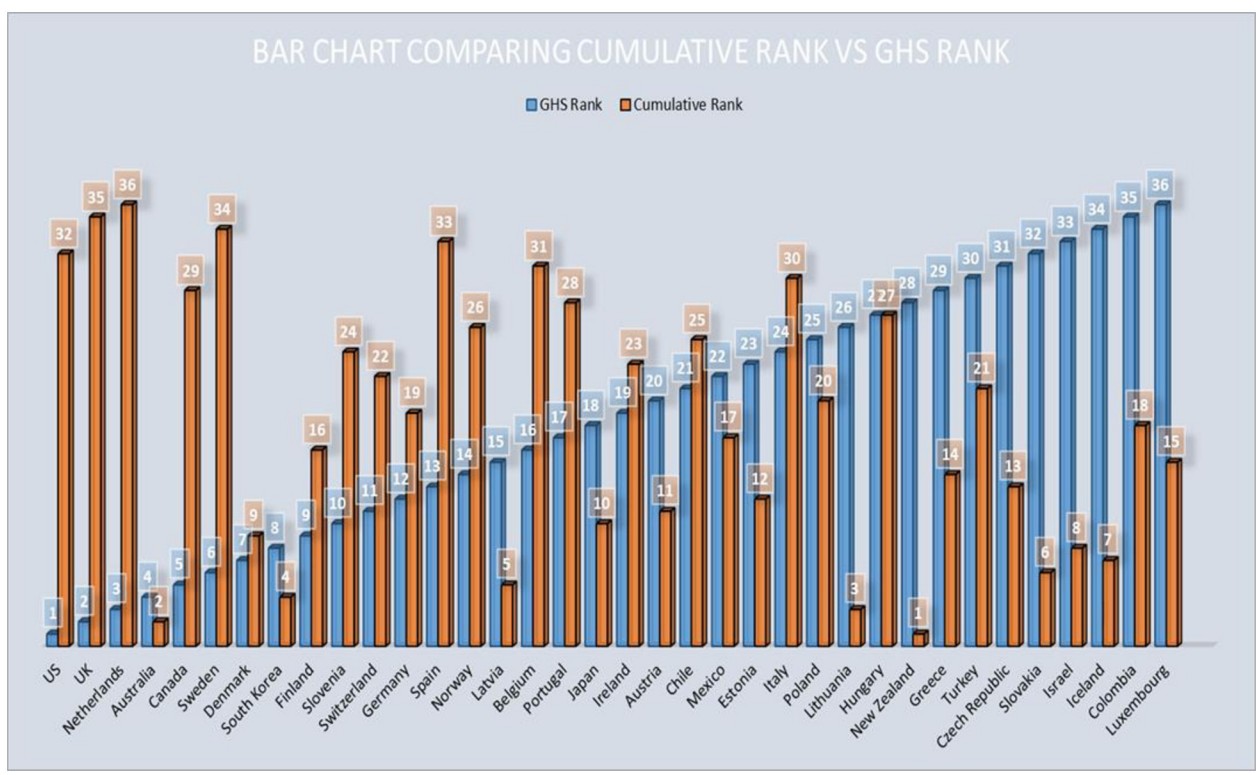

**Fig 1. Comparison of the GHS index ranks with the cumulative rank scores.** GHS = Global Health Security. The above graph represents the OECD countries ranked by the GHS index (shown in blue) in ascending fashion from left to right with a superimposed bar chart (shown in orange), which depicts the cumulative score ranking.

**Table 3. OECD countries and COVID-19 indices.**

| OECD Countries | Total cases/million | Total deaths/million | Recovery Rate (%) | Total tests/thousand | Test Unit |
|---|---|---|---|---|---|
| US | 4283.62 | 259.53 | 17.68 | 32.39 | inconsistent units |
| UK | 3434.45 | 495.15 | 0.45 | 24.5 | people tested |
| Netherlands | 2537.57 | 326.23 | 0.35 | 15.6 | people tested |
| Australia | 274.08 | 3.84 | 90.99 | 38.58 | tests performed |
| Canada | 1944.8 | 144.98 | 50.28 | 32.67 | people tested |
| Sweden | 2830.11 | 349.43 | 17.39 | 17.56 | people tested |
| Denmark | 1849.56 | 92.71 | 85.48 | 63.69 | people tested |
| South Korea | 214.9 | 5.07 | 89.41 | 14.18 | cases tested |
| Finland | 1109.06 | 51.8 | 81.37 | 25.95 | samples tested |
| Slovenia | 704.21 | 49.54 | 18.44 | 33.12 | tests performed |
| Switzerland | 3510.26 | 183.49 | 89.2 | 39.21 | tests performed |
| Germany | 2066.65 | 93.38 | 87.55 | 37.92 | tests performed |
| Spain | 4923.2 | 587.3 | 0.49 | 41.05 | tests performed |
| Norway | 1507.96 | 42.79 | 0.39 | 39.91 | people tested |
| Latvia | 510.02 | 10.07 | 68.81 | 45.29 | tests performed |
| Belgium | 4684.19 | 768.19 | 26.34 | 47.97 | units unclear |
| Portugal | 2777.27 | 116.12 | 11.75 | 62.85 | samples tested |
| Japan | 128.03 | 5.61 | 63.84 | 1.83 | people tested |
| Ireland | 4825.43 | 30.99 | 81.71 | 52.41 | units unclear |
| Austria | 1777.07 | 69.51 | 90.42 | 39.01 | tests performed |
| Chile | 1937.62 | 18.73 | 44.85 | 17.86 | tests performed |
| Mexico | 330.37 | 34.72 | 71.49 | 1.04 | cases tested |
| Estonia | 1325.25 | 46.74 | 52.5 | 51.89 | tests performed |
| Italy | 3689.86 | 518.81 | 53.88 | 47.56 | tests performed |
| Poland | 46543 | 23.33 | 39.27 | 15.23 | samples tested |
| Lithuania | 555.05 | 19.84 | 63.86 | 79.44 | samples tested |
| Hungary | 353.71 | 45.75 | 37.66 | 13.17 | tests performed |
| New Zealand | 238.06 | 4.36 | 124.39 | 46.44 | tests performed |
| Greece | 265.76 | 14.97 | 49.6 | 11.51 | tests performed |
| Turkey | 1716.28 | 47.51 | 73.32 | 18.35 | tests performed |
| Czech Republic | 779.81 | 27.36 | 64.44 | 32.58 | tests performed |
| Slovakia | 270.53 | 4.95 | 76.57 | 24.89 | tests performed |
| Israel | 1915.42 | 30.62 | 75.92 | 57.67 | tests performed |
| Iceland | 5280.59 | 29.3 | 98.89 | 166.16 | samples tested |
| Colombia | 267.48 | 10.32 | 25.42 | 3.6 | samples tested |
| Luxembourg | 6254.23 | 164.54 | 94.05 | 95.83 | people tested |

findings indicate that in reality, countries such as Iceland and Luxembourg outperformed the US. The discrepancy between the GHS rankings and the performance indicators we utilized may be due to the limited emphasis on testing and possibly the versatility of a country's health system, including its reserve capacity.

The ongoing COVID-19 pandemic has shed light on the crucial role leadership plays in crisis management. Some articles cite decisive leadership and coordinated responses as potential game-changers in a country's response to the pandemic, which may not have been considered by the GHS expert panel [16]. The total number of cases, numbers recovered, deaths, and the tests performed are directly influenced by the decisions made by a country's leadership in

mobilizing critical resources and engaging proper stakeholders. The importance of leadership in response to the ongoing pandemic may perhaps explain why New Zealand outperforms all the other OECD member countries, even though it was ranked 35[th] in the GHS index report. Government-driven rapid responses in New Zealand and Australia have accounted for their impactful performance during the pandemic, as evidenced by their social distancing measures, minimization of non-essential services, and their prioritization of a rapid government-led response [17]. This is in sharp contrast to the UK government's approach, described by Scally et al., as "too little, too late, and too flawed," which saw the formation of a counter "independent SAGE" group to advise publicly on the UK's response to COVID-19 [18]. This was to compensate for the lack of independent scientific counsel by the UK's Scientific Advisory Group on Emergencies (SAGE) which is responsible for coordinating the governmental response to national emergencies [18]. The economic trade-off between shutting down borders, businesses and non-essential activities may have contributed to the delay for some countries to adopt such a strategy at the early stages. This appears to be the case for the UK, US, and Canada, which ranked among the most prepared based on the GHS index but represent some of the worst-performing countries in their COVID-19 response based on our indicators. In contrast, countries ranking lower in the GHS index that fared relatively well during the COVID-19 pandemic took swift and effective action to shut down activities associated with a heightened risk of SARS-CoV-2 acquisition [12, 19].

Our analysis has limitations that must be acknowledged. Like any ecological study, our study is subject to ecological fallacy due to country-level data such as total deaths, tests done, and numbers recovered. The cross-sectional nature of our study makes it difficult to draw any causal conclusions. Another limitation of this study is the variability in COVID-19 reporting, especially with regards to testing. In some countries, the lack of testing capacity for the virus can underestimate the exact number of cases. The GHS Index relied entirely on open-source information, either reported by an international entity or published by countries with the intent that, such information should be equally available to the populace. However, this approach failed to engage key stake-holders involved in directing emergency responses in various countries.

Despite these limitations, our study's key strength lies in its ecological nature, allowing us to easily compare the level of preparedness among different countries. Our research agrees with prior studies that found the lack of readiness on many International Health Regulation indices among countries [20, 21]. This study provides insight into how the six categories of predictive indices can be adjusted to reflect the observed pandemic response metrics. We propose that the prevention of the emergence or release of pathogens and the early detection and reporting category should perhaps be more heavily weighed than the other categories.

## Conclusion

Our study underscores the GHS index report conclusion that no country seems fully prepared to tackle an emerging public health emergency threat. The discrepancy between the GHS index ranking and the actual response of countries based on COVID-19 performance indicators likely indicates that the GHS index may have underestimated the level of preparedness of some countries while overestimating that of others. The expert panel should consider reassessing the GHS index frequently, including the potential incorporation of the effect of leadership in subsequent reports, since this appears to have contributed to the successful responses observed in countries like New Zealand and South Korea. Finally, country's response to prior health threats should be incorporated into developing future GHS index reports.

## Acknowledgments

We acknowledge our data sources https://www.ghsindex.org/, https://ourworldindata.org/coronavirus-testing/ and https://github.com/CSSEGISandData/COVID-19.

## Author Contributions

**Conceptualization:** Enoch J. Abbey, Banda A. A. Khalifa, Modupe O. Oduwole, Samuel K. Ayeh, Richard D. Nudotor, Emmanuella L. Salia, Oluwatobi Lasisi.

**Formal analysis:** Enoch J. Abbey, Samuel K. Ayeh, Seth Bennett.

**Methodology:** Enoch J. Abbey, Samuel K. Ayeh, Seth Bennett.

**Project administration:** Emmanuella L. Salia.

**Supervision:** Allison L. Agwu, Petros C. Karakousis.

**Writing – original draft:** Enoch J. Abbey, Banda A. A. Khalifa, Modupe O. Oduwole, Richard D. Nudotor, Emmanuella L. Salia, Oluwatobi Lasisi, Hasiya E. Yusuf.

**Writing – review & editing:** Allison L. Agwu, Petros C. Karakousis.

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
