## [Decision Letter · Decision Letter 0]

28 Aug 2020

PONE-D-20-22950

The Global Health Security Index is not predictive of coronavirus pandemic responses among Organization for Economic Cooperation and Development countries

PLOS ONE

Dear Dr. Karakousis,

Thank you for submitting your manuscript to PLOS ONE. After careful consideration, we feel that it has merit but does not fully meet PLOS ONE’s publication criteria as it currently stands. Therefore, we invite you to submit a revised version of the manuscript that addresses the points raised during the review process.

I am sorry only one reviewer out of many accepted the challenge of reviewing your manuscript. However, I decided to rely on this one review due to the experience of this reviewer who served his country as the Director General of the Ministry of Health and his decades' experience in Emergency preparedness.  This reviewer had only one minor comment that needs to be discussed in your manuscript, wherever  you believe is the correct place.  

We look forward to receiving your revised manuscript.

Kind regards,

Itamar Ashkenazi

Academic Editor

PLOS ONE

Journal Requirements:

Reviewers' comments:

Reviewer's Responses to Questions

**Comments to the Author**

1. Is the manuscript technically sound, and do the data support the conclusions?

Reviewer #1: Yes

2. Has the statistical analysis been performed appropriately and rigorously? 

Reviewer #1: I Don't Know

3. Have the authors made all data underlying the findings in their manuscript fully available?

Reviewer #1: Yes

4. Is the manuscript presented in an intelligible fashion and written in standard English?

Reviewer #1: Yes

5. Review Comments to the Author

Reviewer #1: Please note that the NTI / GHIndex authors have not been in direct contact with key people responsible for Emergency Preparedness in their country in order to make a real assessment.... they just summarized publications that were published.... that a significant fault that causes misleading conclusions about scoring the level of preparedness, effectiveness of preparedness etc.......

6. PLOS authors have the option to publish the peer review history of their article (what does this mean?). If published, this will include your full peer review and any attached files.

Reviewer #1: No

---

## [Author Response · Author response to Decision Letter 0]

3 Sep 2020

Reviewer #1:

Please note that the NTI / GH Index authors have not been in direct contact with key people responsible for Emergency Preparedness in their country in order to make a real assessment.... they just summarized publications that were published.... that a significant fault that causes misleading conclusions about scoring the level of preparedness, effectiveness of preparedness etc.......

Response: We thank the Reviewer for the insightful comment. We agree that the one of the limitations of the GHS Index is the failure to engage key persons responsible for emergency preparedness in their respective countries and other key stakeholders. This point has been discussed in the revised manuscript (lines 187-189).

---

## [Editor Report · Decision Letter 1]

7 Sep 2020

The Global Health Security Index is not predictive of coronavirus pandemic responses among Organization for Economic Cooperation and Development countries

PONE-D-20-22950R1

Dear Dr. Karakousis,

We’re pleased to inform you that your manuscript has been judged scientifically suitable for publication and will be formally accepted for publication once it meets all outstanding technical requirements.

Kind regards,

Itamar Ashkenazi

Academic Editor

PLOS ONE
---

## [Editor Report · Acceptance letter]

16 Sep 2020

PONE-D-20-22950R1

The Global Health Security Index is not predictive of coronavirus pandemic responses among Organization for Economic Cooperation and Development countries

Dear Dr. Karakousis:

I'm pleased to inform you that your manuscript has been deemed suitable for publication in PLOS ONE. Congratulations! Your manuscript is now with our production department.

Kind regards,

on behalf of

Dr. Itamar Ashkenazi 

Academic Editor

PLOS ONE